# Liposomes as Carriers of Bioactive Compounds in Human Nutrition

**DOI:** 10.3390/foods13121814

**Published:** 2024-06-09

**Authors:** Magdalena Rudzińska, Anna Grygier, Geoffrey Knight, Dominik Kmiecik

**Affiliations:** Faculty of Food Science and Nutrition, University of Life Sciences, 60-637 Poznań, Poland; anna.grygier@up.poznan.pl (A.G.); gk.knight91@gmail.com (G.K.); dominik.kmiecik@up.poznan.pl (D.K.)

**Keywords:** liposomes, formation, encapsulation, bioactive compounds, extracts, antimicrobial, food components, stability

## Abstract

This article provides an overview of the literature data on the role of liposomal structures and encapsulated substances in food technology and human nutrition. The paper briefly describes how liposomes are created and how they encapsulate food ingredients, which can either be individual compounds or plant extracts. Another very interesting application of liposomes is their use as antimicrobial carriers to protect food products from spoilage during storage. The encapsulation of food ingredients in liposomes can increase their bioavailability, which is particularly important for compounds with health-promoting properties but low bioavailability. Particular attention was paid to compounds such as phytosterols, which lower blood cholesterol levels but have very low absorption in the human body. In addition, consumer expectations and regulations for liposomes in food are discussed. To date, no in vivo human studies have been conducted to indicate which encapsulation methods give the best results for gastrointestinal effects and which food-added substances are most stable during food storage and processing. The paper identifies further lines of research that are needed before liposomes can be introduced into food.

## 1. Introduction

Encapsulation has been described as a method of temporarily entrapping an active substance like a drug or bioactive compound within a shell-forming material to prevent loss of functional properties and premature degradation [1]. The use of versatile encapsulation means like liposomes as a distribution system for various molecules presents a multidisciplinary challenge, as much in food chemistry as nanomedicine, which has been gaining wider attention. Not only is encapsulation becoming a better understood process but these new alternatives and the settling of the theoretical bases of liposomal application are also making this technology more feasible across both the food and pharmaceutic industries.

Liposomes are a type of nanoparticle to have found widespread use as deliverers for targeted drugs and as a source of functional compounds in food. The basics of their creation, use, and functioning in the human body were first described by Bangham [2], whose work described the modification of the structure of lipids and their biological properties. However, seven years previously, in 1956, Holmes and Moorhouse [3] had described liposomes as droplets containing phospholipids related to granulocytes of the blood. These researchers were seeking a method to distinguish liposomes using colorants. Since then, research into the use of these particles has greatly accelerated, with over 320,000 publications to date (WoS 5.12.2023) describing their structure, use, biological and chemical properties, and their stability in the digestive system, during storage, and in the processing of food products.

Liposomes are mainly used as targeted drug deliverers. They have improved therapies for a range of biomedical applications by stabilizing therapeutic compounds, overcoming obstacles to cellular and tissue uptake, and improving the biodistribution of compounds to target sites in vivo [4]. The use of liposomes in cosmetics is based on the similarity of the bilayer structure of lipid vesicles to that of natural membranes. This includes the ability of lipid vesicles, depending on their lipid composition, to alter the fluidity of the cell membrane and fuse with cells. In this way, they deliver active compounds to the target site not only in the stratum corneum but also in the dermis, in a much shorter time and at higher doses than traditional preparations [5].

The last two decades have seen a strong interest in liposomes in food technology and dietetics. The diet has a significant impact on the functioning of the human body. Despite much advanced biological, medical, and genetic research, there are still many risk factors for diet-related diseases, such as overweight, obesity, type-2 diabetes, cardiovascular disease, and some cancers. By selecting the right ingredients in the diet, and the correct proportions, these risks can be significantly reduced. The presence of health-promoting ingredients in food has been assigned a supporting role in the prevention of certain diseases. Such compounds include nonnutritive substances with documented beneficial effects on the health and performance of the human body, called bioactive compounds. Bioactive ingredients such as carotenoids, polyphenols, antioxidants, vitamins, and phytosterols have been found for a number of commercial and clinical applications. However, these compounds are susceptible to oxidation and degradation during food processing and heating. Encapsulating them within lipid-based nanocarriers makes their application in food products possible.

It is envisaged that liposomes can be health promoting. They can be used to increase the stability of labile compounds, allowing for applications of these sometimes potent biological molecules. Liposomes provide an essential additional administration strategy [6]. They have been described as a delivery system for food supplements [7]. Encapsulation in liposomes overcomes the obstacles that exist with molecules with low stability and poor bioavailability [7]. The literature clearly indicates that liposomal encapsulation is an enhancing strategy that can coexist with the engineering of emulsions, microemulsions, and biopolymer-based systems to derive newer, better formulas in the food field.

Consumer expectations often force food producers to look for new products and manufacturing technologies. In recent years, bioactive compounds with specific biological properties have been of particular interest. Consumers also want food to stay fresh for longer storage periods. Wu et al. [8] points out that the future perspective is to look for composite materials that have the advantages of both antioxidant and antimicrobial activities. Liposomes offer such an option that can combine the two activities, whether as a composite when loaded or embedded with an agent or bioactive component (such as silver nanoparticles, AgNPs) [8]. Compounds encapsulated in liposomes can affect the sensory properties of food products, such as color and aroma. Yogurts enriched in carotenoids and polyphenols extracted from red pepper encapsulated in liposomes have been noted to cause changes in color [9]. The addition of liposomes containing fish oil to yoghurt resulted in consumer scores close to those of the control sample [10].

Researchers are investigating the use of liposome carriers to induce a protective effect on the human body with the natural absorption integration properties of liposomes. Before delving into the focus of this review, liposomes, it is crucial to come to terms with the range of methods that exist to prepare different classes of encapsulation types.

## 2. Formation of Liposomes

The literature describes liposome preparation methods only in general terms, as these methods depend on the class of liposome and type of encapsulation. A simple and direct way of preparing liposomes has been documented by Gibis et al. [11]. A high shear disperser was first applied to a solution, and then a high-pressure homogenizer with a cooling unit was used, yielding the liposomes.

Liu et al. (2020) [12] described both conventional and novel methods of liposome preparation. No knowledge of chemical synthesis is needed, as the process only involves mixing four ingredients, namely, L-α-phosphatidylcholine, cholesterol, Tween-80, and vitamin E in a mass ratio of 6:1:1.8:0.12. These are dissolved in ethanol, and the mixture is dried under a vacuum using a rotary evaporator. The spheroidal dimensional echelon is obtained afterward by rehydrating the lipid strip. Liposomes with lactoferrin instead of cholesterol have been prepared in this way [7]. The effects of cholesterol, β-sitosterol, and stigmasterol on the liposome membrane and their changes during storage have been described [13]. Generally, the addition of sterols enhanced the storage stability of the liposomes. β-Sitosterol was the most efficient with respect to the inhibitory and antioxidant effect of lipid hydrolysis, while cholesterol made the membrane more compact and ordered. Stigmasterol was the less efficient in inducing lipid packing [13].

Wu et al. [8] has reported a unique synthetic preparation of liposomes that is aimed specifically at encapsulating AgNPs. Their method is essentially an application of the thin-film hydration method, which uses lecithin as a model of the liposome bilayer membrane. Lignin was used as a reducer and stabilizer to improve the encapsulation efficiency [8].

Other methods have also been described, including the crystal-film method, modified thin-film deposition, supercritical techniques, and spray drying [14,15,16]. Spray drying is a common technique used to produce proliposomes for food applications.

Apart from the time taken by these various preparation and synthesis methods, a point to note is that made by Toniazzo et al. [17]: methods that are to be scaled up in the future need to be free of any use of organic solvents. Parameter control such as temperature conditions and agitation that enable hydration of phospholipids while avoiding the use of organic solvents are thus ideal and constitute a general desideratum for synthesis methods. Such methods generally use the proliposome approach [17]. Avoidance or reduction in use of organic or toxic solvents is an important aspect of using liposomes as carriers of biologically active substances [18].

Scaling up the production of liposomes for food products presents several technical challenges that need to be addressed for successful commercialization. Liposome production equipment is often customized by manufacturers to meet the specific requirements of different products. This customization involves various mechanical engineering technologies and can be a big challenge [19]. Also, developing reproducible manufacturing protocols, and strategies to minimize leakage and enhance cargo retention, are critical for successful scaling up the production of liposomes [20]. Moreover, the raw material quality, sterility and long-term stability of liposomes, and easy and rapid methods of quality control play important role in scaling up the production of liposomes.

The cost of liposome production can vary based on several factors, including the manufacturing method, scale, and specific application. According to market average selling prices, liposomes are priced at approximately 1.8 EUR/mL. To quickly enter the market, liposomes produced using a lab-scale method for liposome production using supercritical-assisted techniques were priced at 1.1 EUR/mL. The payback time for this business was identified as the fourth year, with a return on investment (ROI) of about 230% and a return on sales (ROS) of 26.7% over a 10-year business plan [21]. But we have to remember that liposome production costs can vary based on individual circumstances and specific business models.

## 3. Encapsulation in Liposomes

One way to prepare loaded liposomes is by encapsulating the ingredient in nanoliposomes by direct injection [7]. Encapsulation is affected by the nature and format of the carrier. With liposomes, the phospholipids take a concentric form, with a hydrophobic shell that protects the hydrophilic aqueous core. Liposome domains can encapsulate hydrophilic and hydrophobic compounds within them [8,22]. Liposomes can serve as carriers for a range of molecule types [23] and, in fact, it is because liposomes can be used as a resource to dissolve both water-soluble and lipid-soluble molecules at the same time that they are used in the food and nutraceutical industries.

According to those in the field, liposome encapsulation and especially nanoencapsulation enhance the efficacy and stability of bioactive substances in food systems. This gives a means to have some control over the release of bioactive compounds or to decrease their loss. The analysis of Wu et al. [8] supports the conclusion that liposomes loaded with AgNPs are regular spheroids with good dispersion and a stable structure. This is conceptualizable directly from their indication that liposomes can be used to encapsulate essential oils and nanoparticles [12].

Among the several ways of synthesizing liposomes, there are two main methods of loading ingredients: passive and active. Passive refers to situations in which the compounds are allowed to incorporate into the liposomes at mixing. Active loading is when the incorporation of the ingredients in the liposomes depends on a specific physicochemical mechanism, such as the presence of electrostatic force between the active molecule and the liposome or the addition of weak acids and bases; both of these are feasible ways of placing bioactives into stable liposome carriers [18].

Encapsulation is determined by total phenolic content analysis, as described elsewhere [24]. Encapsulation efficiency is defined as the ratio of the content of the phenolic compounds expressed as a percentage. Wu et al. [8] also describe the preparation of coated liposomes with low-molecular-weight chitosan.

Liposomes of meat origin have been described by Chen et al. [25]. Minced fillet and pig fat were prepared to a sufficient level to be divided into a control group and a series of liposome–phycobiliprotein samples of increasing load. The samples were sealed with a preservative film for storage; further details are given in the paper [25]. Advancement in liposomes’ use are of value in many fields. For example, liposomal delivery can be used in nematode research, with loads inducing fat accumulation or loss in the nematode [6]. Perhaps surprisingly, the liposome carrier can also support antilipidemic compounds. Modulation of genes has been reported.

Liposomes and other encapsulation methods, such as microemulsions and biopolymer-based systems, serve as versatile vehicles for drug delivery and other applications. They differ in structure, properties, and applicability. Liposomes can encapsulate both hydrophilic and lipophilic compounds, they are biocompatible, and they can be modified for different compounds, but they are very fragile. While they closely resemble eukaryotic cell membranes, their fluidity may not always match that of natural cells [26]. Microemulsions are thermodynamically stable, optically transparent mixtures of oil, water, and surfactants, and form spontaneously without external energy input. They can accommodate high concentrations of hydrophobic compounds and solubilize lipophilic components, aiding their absorption. In microemulsions, a high surfactant concentration is required and their structure is less controllable compared to liposomes [27]. Biopolymer-based systems use natural polymers (e.g., proteins, polysaccharides) to encapsulate compounds. They are biodegradable and environmentally friendly and can be modified for targeted compounds’ delivery. They are rather dedicated for medicine because they allow sustained drug release [26]. The correct choice depends on the specific application and desired properties.

## 4. Liposome Structure and Stability

Liposome evagination is thus dependent on the coating and concentration levels of the embedded species. It is considered that liposomes can take on a range of structures, falling into four groups depending on the size and number of bilayers: small or large unilamellar vesicles, multilamellar vesicles, or multivesicular vesicles [28]. When liposomes have a monophospholipid bilayer in a unilamellar structure, they are referred to as having an onion-like structure. The quantity of encapsulated compounds and their interaction with cell membranes depend on the size of the liposomes and the number of bilayers. Structurally, liposomes have a hydrophobic tail and hydrophilic head and are made from natural or synthetic phospholipids [29]. They are amphiphilic in nature [30]. The fatty acids of the lipid bilayer membrane have significant affection to structure and furthermore encapsulating molecules such as vitamin A, tocopherol, or cholesterol.

Each type of closed molecule has its own encapsulation mechanism, which needs to be known to carry out the liposome preparation protocol with high encapsulation efficiency. The most frequently used and most appropriate method takes lipid film hydration as the first step. Because liposomes mimic natural cell membranes, they have long been investigated as drug carriers with excellent entrapment capacity, biocompatibility, and safety [31,32,33]. Modulating the composition of liposomal lipid bilayers can greatly improve both stability and permeability of the liposome for encapsulation and delivery.

## 5. Stability of Liposomes and Encapsulated Substances in Food Products

The use of liposomes in food has been the subject of many publications that have demonstrated their broad potential as carriers of bioactive compounds. Despite the widespread and consumer-acceptable use of liposomes in cosmetics and pharmaceuticals, their use as food additives has been limited.

The stability of liposomes is an important aspect of maintaining the quality of the substances they encapsulate. We should consider the stability of liposomes from two points of view: the resistance of pure liposomes to external and internal influences and the protection by the liposome layers of the substances encapsulated in them.

The literature suggests that intense research is underway into the formation of liposomes for consumption. There are investigations into phospholipids, sterols, and coating surfaces of various types [34]. A range of modification efforts have been made. In addition to qualitative studies of the stability of liposomes and the bioactive compounds encapsulated within them, in vitro studies have also been conducted, though these are not sufficient: approval for food use requires in vivo testing, which is very costly and time-consuming. To date, no standard method has been developed to produce liposomes for food use.

The structure of liposomes can be altered by uncontrolled degradation and loss of charge—for example, during intense mixing. The stability of liposomes depends on factors such as the composition of the lipid fraction, the pH of the environment, the temperature, and the surface charge. In order to protect liposomes from degradation, various methods of stabilization have been used, including changing the composition of the lipid fraction, modifying the surface, stabilizing the membrane, and adding surfactants and nanoparticles. At the same time, a range of drying techniques, such as freeze-drying and spray-drying, have been used to preserve liposomes [34,35]. A number of modifications have been carried out to increase the stability of liposomes in food during storage. Because the structure of the phospholipids used to form liposomes has the most significant effect on their stability, a comparative study has been carried out of 1,2-distearoyl-sn-glycero-3-phosphocholine and egg sphingomyelin as a source of phospholipids in the structure of liposomes. Analysis of energy, Van der Waals interaction energy, and electrostatic interaction energy has shown that egg sphingomyelin liposomes are more stable than 1,2-distearoyl-sn-glycero-3-phosphocholine [36]. The addition of decanoic acid to liposomes was found to significantly increase stability, while the addition of stearic acid did not show such an effect [37]. Increasing the proportion of saturated fatty acids in the liposome structure by using hydrogenated soy phosphatidylcholine significantly increased their stability compared to soybean phosphatidylcholine and egg yolk phosphatidylcholine [38]. These studies have shown that the method by which liposomal layers and encapsulated components are produced has a significant impact on stability.

The end use of liposomes also determines other requirements related to their stability and resistance to external factors. The stability requirements for liposomes used in cosmetics, drugs, and food are very different: they mainly concern the physicochemical and oxidative transformations of lipid layers during production and storage of the finished products. In the case of food, there are additional issues, such as the exposure to high temperatures during preparation. Aslan et al. [39] has shown that heating of liposomes with different extracts of *Cinnamomum verum*, *Curcuma longa*, *Zingiber officinale*, and *Laurus nobilis* at 50–80 °C for 30 min caused their degradation from 1% to 15%. The authors suggested that the phenolic compounds had created different hardening effects on the liposome structure and could lead to different effects on liposome layers during heating [39]. The temperature of liposomal curcumin decomposition was 70 °C, which is very close to the temperature of free curcumin degradation at 70–80 °C [40]. This is associated with the transition of the phospholipid bilayer of the liposome from the solid phase to the liquid phase.

The choice of technique used for encapsulation of bioactive compounds depends on the specific compound and its application. Researchers still continue to explore innovative techniques of encapsulation to optimize these effects and unlock the full potential of bioactives. The encapsulation methods play a crucial role in improving the stability and bioavailability of bioactive compounds, benefiting both the pharmaceutical and food industries. The choice of wall materials (lipids, synthetic, or natural polymers) significantly influences the stability of encapsulated compounds. Proper selection ensures protection against degradation. Bioavailability is increased by increase the solubility of encapsulated compounds, leading to better absorption in the gastrointestinal tract.

When encapsulated compounds are spray dried, their stability improves by protecting them from environmental factors, and converting them into dry powder allows for better control of their quality. But some compounds may degrade during the drying process. Lyophilization minimizes the exposure to heat and facilities reconstitution upon use but it requires specialized equipment and a long time. Coacervation provides excellent stability due to the protective polymer shell, but it is limited to specific polymers. Emulsification increases bioavailability by increasing the surface and also improves stability by preventing aggregation; however, nanoemulsions need precise formulations. Electrospinning gave us a large surface area for bioavailability but it is a complex process. Various materials, including lipids, synthetic polymers, and natural polymers, can be used as encapsulating agents. The type and amount of wall formers significantly influence the performance of the preparation [41]. Encapsulation methods offer a versatile toolbox for enhancing the properties of bioactive compounds, making them more effective and suitable for various applications.

## 6. Liposomes as Antimicrobial Nanoparticles

The antimicrobial properties of liposomes depend on the encapsulated components, which act on the cells through different mechanisms. These components can react with the bacterial cell walls and destroy the cells, affecting the fluidity and permeability of microbial membranes through the formation of pores and leakage of intracellular components. Some destroy the DNA and RNA of bacteria and interfere with the synthesis of proteins, inhibit the activity of the enzymes needed for bacterial growth, or affect the electron transport and respiration of microorganisms [42,43,44,45].

The antibacterial mechanisms of essential oils have been described by Hyldgaard et al. [46]. There were several sequential inhibition steps in a particular biochemical pathway: the inhibition of enzymes that degrade excreted antimicrobials, interaction of antimicrobials with the cell wall, and interaction with the cell wall and membrane leading to increased uptake of other antimicrobials. However, the lack of detailed knowledge about the components of essential oils means that we have only a superficial understanding of what governs this synergy or antagonism. Terpenes, terpenoids, and phenylpropanoids have been identified in essential oils: such volatile compounds are not stable during storage. The encapsulation of essential oils in liposomes increases their storage stability, centrifugal stability, thermal stability, and antibacterial properties [47].

Table 1 shows examples of applications of liposomes containing antimicrobial substances used in food. Free liposomes do not exhibit such properties, and their effectiveness depends on the quantity and quality of the encapsulated substances.

## 7. Food Components as Liposome Ingredients

In the last decade, over 10,000 scientific papers (WoS 5 December 2023) have been published on the properties of liposomes with various embedded bioactive compounds, and more than 2000 papers have been published in the last year alone. The ability of liposomes to penetrate cell membranes and carry various substances through them has also led to a very high interest in these nanoparticles in food technology. Liposomes loaded with various substances of natural origin are used in medicine and cosmetology in addition to specialized therapeutic drugs. The substances delivered by these liposome often include compounds found in food as natural ingredients, such as curcumin, silibinin, resveratrol, quercetin, β-carotene, and others (Table 2).

Plant extracts are products formed through an extraction and separation process employing plants as raw materials, in which the original components of the plants are generally not altered. In some cases, excipients may be added to ensure the resulting powder or granular product has good fluidity and resistance to moisture absorption. There are also a small number of liquid or oily plant extracts. Plant extracts may be single extracts or component extracts and can be obtained from fruits, vegetables, herbs, and spices using various methods of extraction, depending on the polarity of the extract. Such methods include sonification, heating under reflux, and Soxhlet extraction. Plant extracts can also be prepared by maceration or percolation of fresh green plants or dried powdered plant material in water or organic solvent systems. Other extraction techniques include supercritical-fluid extraction, pressurized-liquid extraction, microwave-assisted extraction, solid-phase extraction, and surfactant-mediated techniques, each of which possesses certain advantages, including using less organic solvent, improvements in sample degradation, elimination of additional sample clean-up and concentration steps prior to chromatographic analysis, improvements in extraction efficiency, selectivity, and kinetics of extraction. Certain aspects of these techniques can also be automated with ease [85].

Plant extracts are added to food products to avail of their antimicrobial and antioxidant properties [86]. The major components of plant extracts include flavonoids, tocols, phenolic compounds, carotenoids, phytosterols, terpenoids, alkaloids, saponins, aromatic acids, tannins, glucosinolates, and essential oils [87]. The use of plant extracts in food products is limited due to their high volatility, low solubility in water, and low stability during storage and production. Encapsulation of plant extracts can increase their stability and also often their bioavailability in the human body [86]. Essential oils from various plants have been successfully incorporated into liposomes. Plasma-enhanced nutmeg essential oil liposomes were used to protect pork meat batters against deterioration during storage [68]. The essential oil of Satureja khuzestanica leaves encapsulated in liposomes has been used to increase the physicochemical and microbiological quality and to extend the shelf life of chilled lamb meat [88]. The essential oil extracted from *Allium sativum* decreased lipid oxidation and spoilage by bacterial groups in vacuum-packed sausages [51]. Liposomes containing essential oil of cinnamon modified by chitosan and sodium alginate showed good antiseptic effects in chilled pork [67]. Liposomes constitute a suitable system for the encapsulation of volatile unstable essential oil constituents, as in the case of the system developed with clove essential oil and eugenol as its main components [65].

The essential oils extracted from plants contain multiple components, each with different physical and chemical properties. The use of pure compounds can thus allow better prediction of the performance, structure, and stability of liposomes. Functional compounds have also been encapsulated in liposomes to protect them from deterioration, but also to increase their bioavailability. These have been used as industrial additions to food products. Liposomes containing curcumin have been used to enrich corn starch and prepare cake batters with a more homogenous yellow color [89]. Liposomes coencapsulating curcumin and resveratrol had a lower particle size, lower polydispersity index, and greater encapsulation efficiency [79]. When liposomes with limonene were used to coat fresh strawberries, the shelf life of the fruit increased without any decrease in quality [90]. The physical, rheological, and humectability properties of liposomes encapsulating rutin on the sliced surfaces of almonds and chocolate have been investigated and were found to decrease the value of the cohesive energy of almonds and chocolate [91]. Canthaxanthin- and α-tocopherol-loaded liposomes have been used in fish diets, and the effects on fish growth, color, nutrition, and canthaxanthin deposition in the fillets of cultured rainbow trout (*Oncorhynchus mykiss*) were determined [82]. The storage stability and antioxidant activity of quercetin encapsulated in liposomes were better than those of native quercetin, and the inhibitory effect of the Q-NPs on HepG2 cells was comparable with that of free quercetin [92]. The loadability of various carotenoids into the liposomal membrane, the lipid peroxidation inhibition capacity, the storage stability, and the in vitro release behavior in simulated gastrointestinal media were described by Tan et al. [93], who found that carotenoids exhibited various incorporating abilities into liposomes ranging from the strongest to the weakest as follows: lutein > β-carotene > lycopene > canthaxanthin. Liposomes with coencapsulated β-carotene and vitamin E exhibited higher antioxidant activity and bioavailability [70]. Vitamin C was chosen to produce coencapsulated liposomes in order to improve the stability of β-carotene [12], and it was found to improve storage stability and provide an effective delivery system for hydrophilic and hydrophobic compounds. Berberine-loaded liposomes were generated to enhance oral bioavailability and the therapeutic effect [94]. Three other alkaloids (vincristine, vinorelbine, and vinblastine) were encapsulated in liposomes and a comparative study of their loading and retention properties was performed [95]. Sea cucumber saponin liposomes were found to ameliorate obesity-induced inflammation and insulin resistance [62]. Liposomes are good carriers of protein structures, including both amino acids and enzymes or polypeptides. Isoleucine and proline are antihypertensive bioactive peptides with high reactivity, instability in food products, bitterness, and low bioavailability. Loading isoleucine–proline–proline into liposomes added to model functional beverages improved their health-promoting properties, palatability, shelf stability, and bioavailability [96].

Bioactive peptides obtained from the hydrolysis of sheep whey by the bacterial protease were encapsulated into liposomes and were found to be stable after thirty days of storage, which suggests their use as functional foods [97]. Amino acids, peptides, and proteins have been incorporated into liposomes, particularly as antioxidant compounds. Phycobiliproteins are hydrophilic pigment proteins that have great potential in the food industry as functional ingredients. Their use in a liposome system was investigated by X. Chen et al. [25]. Soy protein hydrolysates in liposomal systems were not only able to inhibit the formation of lipid oxidation products but could also interact directly with the liposomal membrane, imparting a positive influence on the stability of the liposomal system [60].

Phytosterols are specific compounds that can be incorporated into liposomes in two ways and which can contain different functions, either replacing cholesterol in the liposomal membrane or being incorporated into liposomes as functional compounds that lower the cholesterol level in the human body. Liposomal membranes containing β-sitosterol were more fluid than bilayers with cholesterol and reduced the average diameter of liposomes [98]. Additionally, liposomes encapsulating β-sitosterol were smaller than their counterparts containing cholesterol due to the higher mobility and deformability of the bilayer membrane. Phytosterols are added as functional compounds to many food products, such as margarines, yoghurts, vegetable juices, cold cuts, and chocolates. They decrease the cholesterol level in human blood, but their absorption is limited, with less than 5% of plant sterols and less than 0.5% of stanols being absorbed and entering the systemic circulation, compared to about 50–60% of dietary cholesterol [99]. It is thus worth investigating and developing the use of phytosterols in liposomes, which are highly effective lipid transporters. In our laboratory, we have prepared liposomes with encapsulated phytosterols and their esters and have determined their physicochemical and biological properties relating to bioavailability in the gastrointestinal tract, the thermo-oxidative stability, cytotoxicity, and genotoxicity (data not published).

## 8. Liposome Applications in Food Products

Many published works describe testing the quality of liposomes enriched with bioactive substances extracted from raw materials or food products (Table 3). While liposomes have found broad applications in pharmacy and cosmetics, the number of publications on their use as functional additives in food products is much more modest. Wheat bread with the addition of liposome-encapsulated garlic extract has been found to be microbiologically more stable than controls, showing mold inhibition for five days. Liposomes formulated with garlic extract thus have potential as a natural antifungal agent in bakery products [63]. Liposomal-encapsulated ethanolic coconut husk extract played a role in an effective hurdle approach for shelf-life extension of Asian sea bass slices, but did not show antimicrobial properties [100].

Polyunsaturated fatty acids (PUFAs) have very important health benefits but are very unstable and have a strong odor, making them good subjects for liposome encapsulation. The stability of PUFA liposomes was determined and, depending on the preparation method, no significant changes were observed over ten months of cold storage (4 °C) in the dark [101]. Oil from shrimp cephalothorax, a by-product of the shrimp processing industry, is a rich source of omega-3 fatty acids and astaxanthin. The encapsulation of shrimp oil in nanoliposomes is a promising technology, which can be utilized to protect the oil against oxidation and mask off-odors. Shrimp-oil liposome powder could be used to fortify a number of food products, particularly beverages. Moreover, the powdered form increases shelf stability, facilitates handling and transportation, and significantly reduces cost [102]. Nanoencapsulated fish oil with a high PUFA content has also been used in functional bread [103]. Encapsulation of mullet fish oil was shown to be effective in protecting this oil from oxidative changes and affecting the protection of rats from titanium dioxide NPs’ genotoxicity [104]. Milk chocolate fortified with polyunsaturated omega-3 and omega-6 fatty acids is a very unstable product, best stored at room temperature. Added antioxidants are usually not stable during heating, a crucial technological process in chocolate production. Milk chocolate fortified with liposomes containing vitamin C and vitamin E as antioxidants was stable after heat treatment and the fortification did not affect their organoleptic properties [71].

The antimicrobial properties of liposome-encapsulated substances have been described in several publications. Some of them have been used as components of coated packaging material.

Composite nanoliposomes encapsulating laurel essential oil and AgNPs were prepared using the film hydration method [8]. The antioxidant properties of the resulting liposomes were examined and found to have good antioxidant activity capable of preventing the oxidative rancidity of food. Furthermore, the prepared packaging films showed excellent antimicrobial activity against *E. coli* and *Streptococcus aureus* and effectively extended the storage life of pork. Chrysanthemum oil and clove oil encapsulated in liposomes have been added to chicken or vegetables as antimicrobial constituents [48,50]. Liposomes encapsulating chitosan and pectin-modified chrysanthemum essential oil possessed high antibacterial activity against *Campylobacter jejuni* on chicken during fourteen days of storage at 4–37 °C, with no effect on the quality of the chicken [50]. The effect of clove oil encapsulated in liposomes on the *E. coli* biofilm exerted a long-lasting antibacterial effect, effectively removing *E. coli* biofilm from the vegetable surface and prolonging shelf life [48]. The applications of liposomes in cheese technology related to the acceleration of cheese ripening by liposomes with enzymes, fortification with vitamins, and increasing the shelf life of cheese using nisin encapsulated in liposomes [105]. Kheadr et al. [106] investigated the effects of liposome-encapsulated bacterial and fungal protease in the ripening of Cheddar cheese and found that the cheese that contained the liposomal enzyme had a higher moisture content and less protein than the controls. Cheese containing nisin encapsulated in liposomes did not show any changes in proteolysis or rheology, but it saw an increase in cheese flavor quality and a reduced bitter taste [107].

Liposomes, which are lipid-based vesicles, have gained significant attention as a versatile encapsulation method for various bioactive compounds. There are some reasons why liposomes are particularly effective compared to other encapsulation methods. They can encapsulate both hydrophobic and hydrophilic molecules. Their lipid bilayer structure allows them to accommodate a wide range of compounds, making them suitable for different applications. Surface modifications allow them to selectively interact with specific cells or tissues, ensuring precise drug or nutrient delivery. Liposomes provide stability to sensitive compounds and they enable controlled release of encapsulated compounds [108]. Liposomes improve the bioavailability of encapsulated compounds and can also encapsulate proteins. Loading proteins into liposomes enhances their stability and bioavailability, making them useful for therapeutic purposes [109]. Because of that, liposomes offer a versatile platform for encapsulating bioactive compounds, making them an attractive component in food.

Liposomes are appearing more and more frequently in food, as carriers of bioactive or a protective (antimicrobial) substances, but their applications are still sporadic. The utilization of liposomes in food is a natural step, after their developments in the pharmaceutical industry and in the production of dietary supplements. In the case of dietary supplements, two types of food-grade ingredients are used to create carriers of bioactive substances: lipophilic substances and surface-active substances. Among lipophilic substances, the following can be used: triacylglycerol oils (canola, corn, fish, medium-chain triglycerides, palm, peanut, soybean, sunflower oils), essential oils (carvacrol, lemon grass, oregano, thyme, thymol oils), flavor oils (lemon, lime, orange, peppermint oils), and indigestible oils (waxes, hydrocarbon, paraffin, mineral oils). The group of surface-active substances includes small-molecule surfactants (monoglycerides, diglycerides, Tweens, Spans, sugar esters), phospholipids (lecithin and lysolecithin), proteins (casein, gelatin, soy, and whey), polysaccharides (arabic gum and modified starch), and solid particles (silica or titanium) [110]. These compounds are commonly used in supplements and are generally available for food production. They are considered safe and do not compromise consumer health. Due to the structure of lipid nanoparticles, the safety of liposomes in food should also be considered in relation to the individual molecules that make up their structure. The presence of triacylglycerols or phospholipids in liposomes leads to the presence of unsaturated fatty acids in their structure [111]. Unsaturated fatty acids undergo oxidation under the influence of atmospheric oxygen and other factors, resulting in the formation of lipid oxidation products (e.g., hydroperoxides, epoxides, aldehydes) [112]. In addition to fatty acids, cholesterol and phytosterols are also used in the formation and stabilization of liposomes for food applications [7,13]. The presence of unsaturated bonds in the structure of sterols leads to their oxidation and the formation of sterol oxidation products, such as 7- or 25-hydroxysterols, 7-ketosterols, epoxysterols, or triols [113].

The presence of lipid oxidation products in the human diet poses a health risk due to their cytotoxic and pro-inflammatory properties. Consumption of lipid oxidation products has been linked to the occurrence of various diseases, including atherosclerosis, Alzheimer’s disease, age-related macular degeneration, diabetes, chronic inflammatory bowel disease, and some forms of cancers [114,115,116,117].

The oxidation process of unsaturated compounds depends not only on the presence of oxygen but also on the temperature and time. The higher the process temperature and the longer the time of heating, the higher the content of oxidized compounds. These are further elements that should be taken into account when assessing the impact of liposomes used in food. Drying is one way of preserving the resulting liposomes prior to their potential food applications. On the other hand, food enriched with liposomes will be subjected to high temperatures during production and heat treatment processes in restaurants or at home.

**Table 3 foods-13-01814-t003:** Food products enriched with bioactive compounds encapsulated in liposomes.

Food Products Enriched with Liposomes	Compounds Encapsulated in Liposomes	Functions of Liposomes	References
Wheat bread	Garlic extract	Antifungal	[63]
Sausages	Garlic essential oil	Antimicrobial, antioxidant	[51]
Asian sea bass slices	Coconut husk extract	Antimicrobial, shelf-life extension	[100]
Functional bread	Fish oil	Health benefits	[103]
Milk chocolate	Vitamin E, vitamin C	Antioxidants	[71]
Vegetables	Extract of clove oil	Antimicrobial	[48]
Chicken	Chrysanthemum essential oil	Antimicrobial	[50]
Pork	Laurel essential oil	Antioxidant	[8]
Orange juice	Resveratrol	Stability of liposomes in orange juice	[118]
Cheese	Enzymatic cocktail	Accelerates proteolysis, lipolysis, and flavor formation	[106]
Cheese	Nisin	Increases flavor, decreases bitter taste	[107]

## 9. Consumer Expectations

Consumer decisions and expectations are sometimes very difficult to determine and may depend on a great many objective and subjective factors. However, the consumer’s point of view is crucial when launching new products. Price is one of the most important factors for consumer attitudes, while taste, convenience, and health properties are playing an increasingly important role [119]. The age and education of consumers affect their purchasing decisions. Liposomes have attracted attention in the food industry on account of their potential applications: they can encapsulate bioactive compounds such as vitamins, enzymes, and polyphenols; they are capable of protecting nutrients against degradation; and they have been used as flavoring compounds and to preserve meat products. They hold great potential for enhancing food quality, safety, and nutrient delivery. In addition, for the growing number of consumers interested in environmental protection, food waste, and waste-free production, they represent attractive ways to manage by-products and waste in practice [120].

Most consumers are afraid of consuming something new that they do not know. Therefore, the most important role in introducing liposomes into food is education. The safety of food additives used in the form of liposomes and their unique properties should be clearly indicated. In addition, regulations should be introduced on the use of liposomes and the substances encapsulated in them in food, with quality and technological requirements that guarantee the safety of these products for consumers [111].

Manufacturers should effectively communicate the benefits and safety of liposome-enriched foods by emphasizing nutrient delivery, safety, controlled release, masking flavors, targeted delivery, longer shelf life, and transparent education. By doing so, they can build consumer trust and encourage the adoption of these innovative food products [108,121].

## 10. Regulation of Liposomes in Food

Nanoparticles, including liposomes, can be used as dietary supplements or ingredients to enrich prepared foods with bioactive substances. However, the safety of their use has not yet been clearly classified. Even if a material is considered safe, its physical and chemical properties on the nanoscale can be quite different. Taking nanoparticles orally can affect the gut microbiota [122] and the regularity of the digestive system [123], and these substances can accumulate in tissues and cells [124,125]. Furthermore, nanotechnology may increase the oral bioavailability of hydrophobic bioactive compounds, causing adverse consequences or health problems in certain cases [126].

Liposomes are not specifically regulated as a separate category in food. Instead, their use falls under broader regulations related to food additives, food ingredients, and safety assessments. Regulatory agencies, such as the European Food Safety Authority (EFSA) in the European Union, evaluate the safety of novel food ingredients, including liposomes. The lipids and phospholipids used in liposomes are themselves subject to safety assessments.

Liposomes used in food should comply with general food safety standards, labeling requirements, and good manufacturing practices (GMPs).

In Europe, the use of nanoparticles in food is regulated by the Novel Foods Directive 2015/2283, which defines ‘nanomaterial’ as a material of special origin that has one or more dimensions less than or equal to 100 nm. These dimensions refer to functional parts enclosed within or held on the surface. When this definition is met, the EFSA assesses toxicological risks in in vivo and in vitro systems [127].

In February 2024, the EFSA published guidance on the scientific principles and data required to assess the safety and bioavailability of new substances proposed as sources of micronutrients for use in food supplements, fortified foods, and foods for specific groups of consumers [128]. In summary, the EFSA is actively working to ensure the safety of food supplements, including those containing liposomes.

Regulatory bodies need more research to evaluate safety of liposomes before their widespread use in food products [111]. Regulatory agencies require evidence that liposomes do not degrade or release their contents prematurely, affecting product quality and safety [129]. Clear labeling is essential to inform consumers about the presence of liposomes in food products. Regulations may require manufacturers to disclose the use of liposomal technology on product labels, and standardization protocols are needed for liposome preparation and characterization. The bioavailability of liposome-encapsulated ingredients has to be regulated. Compatibility studies should evaluate how liposomes behave when combined with various food matrices. If liposome-encapsulated ingredients are considered novel foods, regulatory approval may be necessary before their introduction into the market. Manufacturers must provide safety data and demonstrate the technological need for using liposomes in specific food applications.

Regulatory bodies worldwide are actively engaged in assessing and managing the use of nanotechnology in food applications to ensure safety, quality, and compliance with existing standards. The United States Food and Drug Administration (USFDA) evaluates the safety and efficacy of nanomaterials used in food products and ensures compliance with existing regulations. Food Standards Australia and New Zealand (FSANZ) also actively participates in ensuring the safety and proper use of nanotechnology in food products [130]. Moreover, the Organization for Economic Co-operation and Development (OECD) works to provide valuable insights into global regulatory practices [131].

## 11. Conclusions

The main purpose of this review has been to provide an overview of the use of liposomes in food, both as carriers of nutrients and as ingredients in food products. One very interesting thread is the use of liposome films to protect food products from spoilage during storage.

The last two decades have seen a surge in publications on liposomes and their structures, preparation, modifications, and use, mainly in medicine. There have been in that period about 10,000 works on liposomes as drug carriers, while their use in food has been described in about 150 publications. These papers have mainly studied the stability of various functional food components encapsulated in liposomes during production and storage, and their bioavailability in the human body. They represent a very promising means of delivering bioactive compounds with a traditional meal. The hydrophilic–hydrophobic nature of liposomes opens up a wide range of possibilities of use for both researchers and industry.

Particular attention should be paid to the role of cholesterol and plant sterols not only as structural components of liposomes but also as components encapsulated in liposomes. Protecting phytosterols from oxidation and enhancing their uptake in the human body may prove to be of key importance in preventing cardiovascular heart disease and hormone-dependent cancers.

The most critical areas for future research in liposome technology for food applications are maximizing the benefits and reducing the risks of degradation during food storage, processing, and digestion. Liposome technology offers exciting opportunities for improving food quality, nutrient delivery, and overall health, but their stability, bioavailability, toxicity, and some other properties must be analyzed in detail. Consumer acceptance also plays an important role in the use of liposomes in food.

A crucial factor in advancing the development and application of liposomes in nutrition is the interdisciplinary approaches adopted. Collaboration between scientists from different disciplines such us food science, biochemistry, pharmacology, dietetics, diagnostics, and biomedical engineering fosters innovation, accelerates research, and ensures that liposomes become valuable tools for enhancing nutritional delivery. By combining expertise from various fields, we can unlock the full potential of liposomal technology in improving health and well-being.

However, the further use of liposomes in food products still requires much research. It will be necessary to develop technology for the large-scale production of liposomes and to obtain bioactive substances with health-promoting properties for encapsulation in these nanostructures. Extracts made from food industry by-products rich in bioactive compounds could find applications as enrichments of traditional foods such as bread, milk, yoghurt, and meat products.

## Figures and Tables

**Table 1 foods-13-01814-t001:** Antimicrobial liposomes in food products.

Encapsulated Compounds	Food Products	Antimicrobial Properties	Liposomes Used and Mechanisms of Antimicrobial Interactions	References
Clove oil	Vegetable surfaces	*Escherichia coli*	Bacterial concentration was 10^3^–10^4^ CFU/mL; samples were incubated at 25 °C for 48 h and immersed in 5 mg/L and 10 mg/L clove oil liposome solutions	[48]
Herb oils (rosemary, laurel, thyme, and sage)	Rainbow trout	Mesophilic aerobic bacteria and psychrophilic viable counts	Liposomes were made of water, Tween 60, soybean oil, glycerol monooleate, refined soya sterols, and the cationic compound cetylpyridinium chloride. Nanoemulsions with the liposomes were then prepared and fish fillets were immersed for 3 min in the nanoemulsions	[49]
Chrysanthemum essential oil	Stored chicken	*Campylobacter jejuni*	Bacterial concentration was 10^3^ CFU/g; chicken samples were sprayed with liposomes at 1 mL/100 g	[50]
Garlic essential oil	Sausages	Spoilage bacterialgroups	The efficacy of chitosan and whey protein films impregnated with garlic essential oil (GEO, 2% *v*/*v*) or nanoencapsulated GEO was assessed for their ability to extend the shelf life of refrigerated vacuum-packed sausages; the comparison was made over fifty days	[51]
Essential oils from carvacrol, thymol, eugenol, trans-cinnamaldehyde, β-resorcylic acid, and vanillin	Soy sauce	*Escherichia coli*, *Salmonella typhimurium*, *Listeria monocytogenes*	The membrane attacking properties of carvacrol and thymol	[52]
Basil essential oil	A novel unidirectional water-conducting sustained-release preservation pad for fish fillets	*Shewanella putrefaciens*, *Pseudomonas fluorescens*	The loading efficiency and loading capacity of the liposomes were 90% and 50%, respectively. Basil essential oil liposomes were used in unidirectional water-conducting preservation pads; this provided the antibacterial properties, effectively inhibiting the growth of microorganisms and extending the shelf life of L. Japonicus fillets.	[47]
Leaf extract of Rosmarinus officinalis L.	In vitro antimicrobial activity against all selected bacterial reference strains	Gram-positive species (*Bacillus cereus*, *Enterococcus faecalis*), Gram-negative species (*Salmonella enterica*, *Escherichia coli*)	Liposomes loaded with R. officinalis leaf extract were prepared using a modified version of the reverse-phase evaporation technique. R. officinalis extract (a volume 50% higher in relation to the final liposomal dispersion) and distilled water were added to the ethanolic solution. The minimum inhibitory and bactericidal concentrations established using the broth microdilution method indicated better antimicrobial activity against Gram-positive strains (MIC/MBC ≤ 4)	[53]
Tea polyphenols	Aquatic products	*Shewanella putrefaciens*, *Pseudomonas fluorescens*	Bacterial membrane fluidity was reduced and the cell membrane structure was damaged. Polyphenols are released from the liposomes and can damage the bacterial cell membrane leading to a cavity in the bacterial cell. Intracellular substances such as DNA and ATP enzyme are leaked	[54]
Cinnamaldehyde	Bacterial biofilm on stainless steel	*Listeria monocytogenes*, *Salmonella enteritidis*	MIC = 0.625–1.25 µL mL^−1^ MBC = 2.5–5.0 µL mL^−1^	[55]
Nisin	Whole and skim milk	*Listeria monocytogenes*	Liposomes consisting of distearoylphosphatidylcholine and distearoyl phosphatidylglycerol, with 0, 5, or 10 μg/mL of the antimicrobial peptide nisin entrapped, were exposed to elevated temperatures (25–75 °C) and a range of pH values (5.5–11.0). Encapsulation efficiencies were approximately 89–91%, 78–83%, and 72–78% for PC/PG 6:4, PC/PG 8:2, and PC, respectively, at pH = 5.5–11.5	[56,57]

**Table 2 foods-13-01814-t002:** Extracts of food components encapsulated in liposomes.

Bioactive Compounds	References
Multicompound extracts	
Black carrot extract	[15]
Grape seed extract	[58]
*Moringa oleifera* L. extract	[59]
Protein isolate hydrolysates	[60]
Tomato-peel extract	[61]
Sea cucumber saponins	[62]
Garlic extract	[63]
Coconut husk	[64]
Essential oils	
Laurel essential oil	[8]
Clove essential oil	[65]
Lemongrass oil	[66]
Garlic essential oil	[51]
Cinnamon essential oil	[67]
Nutmeg essential oil	[68]
Single-compound extracts	
Plant sterols	[13,69]
Betalain	[6]
β-Carotene	[12,17,70]
Vitamin C	[12,71]
Polyphenols	[16]
CLA isomers	[23]
Resveratrol, quercetin	[22,72]
*Monascus* red pigment	[73]
Triterpenols	[74]
Cinnamaldehyde	[75]
Silibinin	[76]
Acteoside	[77]
Curcumin	[78,79]
Alcalase hydrolysate	[80]
Rosmarinic acid	[81]
Canthaxanthin	[82]
*α*-Tocopherol	[70,71,82]
Nisin (Bacteriocin)	[83,84]

## Data Availability

The original contributions presented in the study are included in the article, further inquiries can be directed to the corresponding author.

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
