# Peer review of "Liposomes as Carriers of Bioactive Compounds in Human Nutrition"

_foods, 2024, doi:10.3390/foods13121814_

Round 1

Reviewer 1 Report (Previous Reviewer 3)

Comments and Suggestions for Authors

The present review manuscript "Liposomes as Carriers of Bioactive Compounds in Human Nutrition" provided by the authors explores the potential of liposomes as carriers for bioactive compounds in food science. The review covers the possibility of formation, encapsulation, structural formation, and stability of liposomes, as well as their other virgin applications as drugs, cosmetics, and food. Throughout, the authors stress the potential of liposomes to improve the bioavailability, stability and antimicrobial activity of different food components and ingredients, as well as considering consumer expectation and regulatory aspects.

In my opinion, the authors have made a significant contribution to the scientific community's understanding of this subject. The review manuscript is well-written and offers valuable information. However, I would like to provide some comments that I believe would further enhance the manuscript's quality before possible consideration for publication in the Journal as detailed in the following:

Author Response

Thank you very much for taking the time to review this manuscript. Please find the detailed responses below and the corresponding corrections highlighted in yellow color in the re-submitted manuscript. 

Reviewer 2 Report (New Reviewer)

Comments and Suggestions for Authors

Liposome is one of the important lipid-based delivery systems and have been widely used in cosmetics and pharmaceuticals. However, the use of liposomes in food as an encapsulation and delivery system of bioactive compound or antimicrobial agents is a new direction. This manuscript just reviewed the liposome as carriers of bioactive compounds in food system comprehensively, and I think it can be accepted for publication after a revision.

1. this manuscript mainly focus on the application of liposomes in food, so these contents in cosmetics and drugs can be removed, since there are many reviews published.

2. line 11, ‘we cover the use of liposomes in cosmetics and drugs, which is their major application’ can be deleted, the same with ‘5. Applications of liposomes in cosmetics and drug’.

3. line 116, please pay attention to ‘[17]: methods that……’, is the format correct?

4. ‘7. Liposomes as antimicrobial nanoparticles’, why is this part introduced separately? If liposomes as antimicrobial carriers for food preservation but the content is not in-depth.

5. Conclusions and prospects are better presented in terms of items.

Author Response

Thank you very much for taking the time to review this manuscript. Please find the detailed responses below. 

Round 2

Reviewer 1 Report (Previous Reviewer 3)

Comments and Suggestions for Authors

The manuscript is well-written, logically structured, and easy to follow, making it accessible to a broad audience.

Overall, this manuscript makes a significant contribution to the field and will be of great interest to the readers.

This manuscript is a resubmission of an earlier submission. The following is a list of the peer review reports and author responses from that submission.

Round 1

Reviewer 1 Report

Comments and Suggestions for Authors

The paper titled "Liposomes As a Source of Bioactive Compounds in Human Nutrition" explores the use of liposomes in food technology, emphasizing their role as carriers of nutrients and protective agents against food spoilage. While the paper presents a comprehensive overview of liposome applications in food and related industries, several aspects could be improved, especially concerning the microbiological approach to using liposomes in food systems:

1.       The paper discusses the antimicrobial properties of liposome-encapsulated compounds like garlic extract, laurel essential oil, and AgNPs. However, it lacks specificity regarding the spectrum of microbial inhibition (e.g., specific bacteria or fungi targeted), the concentration of liposomes needed for effective antimicrobial activity, and the mechanism of action on microbial cells​​. Please, add this in the paper.

2.       The paper mentions various methods for liposome preparation and encapsulation techniques but lacks standardization of these formulations for specific food applications. Standardization is crucial for ensuring consistent antimicrobial effectiveness and safety in food products​​. Please, add this in the paper.

3.       While the paper covers the use of liposomes for extending the shelf life of food products, it lacks detailed studies or data on the stability of these liposomal formulations over time, especially under different storage conditions relevant to food products​​. Please, add this in the paper.

4.       There is minimal discussion on the safety and regulatory aspects of using liposomes in food products. Since food safety is paramount, addressing potential toxicity, allergenicity, and regulatory compliance of liposomal ingredients is crucial​​. Please, add this in the paper.

5.       The paper could be strengthened by addressing the challenges of scaling up liposome production for industrial applications in the food industry. This includes aspects like cost-effectiveness, manufacturing practices, and maintaining product quality during large-scale production​​. Please, add this in the paper.

6.       The paper does not address consumer perception and acceptability of liposome-incorporated food products. Understanding consumer attitudes towards such novel food technologies is critical for successful market introduction. Please, add this in the paper.

7.       A comparative analysis of liposome-based preservation methods with traditional or other novel food preservation techniques would provide a clearer understanding of the relative advantages and disadvantages. Please, add this in the paper.

8.       More detailed microbiological aspect and posibility for encapsulation of of certain microorganisms such as probitics in liposomal structures, and studies on the interaction of liposomes with microbial cells would enhance the paper's depth and relevance to food microbiology. Please, add this in the paper.

Comments on the Quality of English Language

Moderate editing of English language required

Reviewer 2 Report

Comments and Suggestions for Authors

The work is an interesting review regarding liposome preparation and application. Focusing on food products enriched with bioactive compounds encapsulated in liposomes is a nice strategy. The paper could be interesting for the audience in Foods. I am concerned about the title (Liposomes As a Source of Bioactive Compounds in Human Nutrition) because liposomes are not a source of bioactive compounds. Maybe “Liposomes As Carriers of Bioactive Compounds in Human Nutrition” would work better. 

Reviewer 3 Report

Comments and Suggestions for Authors

This contribution aims to provide an overview of the use of liposomes in food, both as carriers of nutrients and as ingredients in food products. Another very interesting application is the use of liposome films to protect food products from spoilage during storage. Authors present some general information on the formation of liposomes and the mechanisms of different encapsulated compounds and provide an outline of liposome structures. the review covers the use of liposomes in cosmetics and drugs, which is their major application. Liposomes are however increasingly finding application in food. Two directions of use can be distinguished here: liposomes as carriers of extracts or individual food components for use in dietary supplements or cosmetics, and liposomes as food additives. The encapsulation of food components in liposomes can increase their bioavailability, which is particularly important for compounds that have health-promoting properties but low absorption.
